# Cancerous and Non-Cancerous MRI Classification Using Dual DCNN Approach

**DOI:** 10.3390/bioengineering11050410

**Published:** 2024-04-23

**Authors:** Zubair Saeed, Othmane Bouhali, Jim Xiuquan Ji, Rabih Hammoud, Noora Al-Hammadi, Souha Aouadi, Tarraf Torfeh

**Affiliations:** 1Department of Electrical & Computer Engineering, Texas A&M University, College Station, TX 47080, USA; jim.ji@qatar.tamu.edu; 2Department of Electrical & Computer Engineering, Texas A&M University at Qatar, Doha 23874, Qatar; othmane.bouhali@qatar.tamu.edu; 3Department of Science & Arts, Texas A&M University at Qatar, Doha 23874, Qatar; 4Qatar Center for Quantum Computing, College of Science and Engineering, Hamad Bin Khalifa University, Doha 34110, Qatar; 5Department of Radiation Oncology, National Center for Cancer Care and Research, Hamad Medical Corporation, Doha 3050, Qatar; rhammoud2@hamad.qa (R.H.); nalhammadi1@hamad.qa (N.A.-H.); saouadi@hamad.qa (S.A.); ttorfeh@hamad.qa (T.T.)

**Keywords:** deep learning, magnetic resonance imaging, classification, machine learning, deep convolutional neural networks, dual DCNN model, inceptionV3, denseNet121, resNet50, resNet34, resNet18, efficientNetB2, squeezeNet, VGG16, alexNet, leNet-5, learning rates

## Abstract

Brain cancer is a life-threatening disease requiring close attention. Early and accurate diagnosis using non-invasive medical imaging is critical for successful treatment and patient survival. However, manual diagnosis by radiologist experts is time-consuming and has limitations in processing large datasets efficiently. Therefore, efficient systems capable of analyzing vast amounts of medical data for early tumor detection are urgently needed. Deep learning (DL) with deep convolutional neural networks (DCNNs) emerges as a promising tool for understanding diseases like brain cancer through medical imaging modalities, especially MRI, which provides detailed soft tissue contrast for visualizing tumors and organs. DL techniques have become more and more popular in current research on brain tumor detection. Unlike traditional machine learning methods requiring manual feature extraction, DL models are adept at handling complex data like MRIs and excel in classification tasks, making them well-suited for medical image analysis applications. This study presents a novel Dual DCNN model that can accurately classify cancerous and non-cancerous MRI samples. Our Dual DCNN model uses two well-performed DL models, i.e., inceptionV3 and denseNet121. Features are extracted from these models by appending a global max pooling layer. The extracted features are then utilized to train the model with the addition of five fully connected layers and finally accurately classify MRI samples as cancerous or non-cancerous. The fully connected layers are retrained to learn the extracted features for better accuracy. The technique achieves 99%, 99%, 98%, and 99% of accuracy, precision, recall, and f1-scores, respectively. Furthermore, this study compares the Dual DCNN’s performance against various well-known DL models, including DenseNet121, InceptionV3, ResNet architectures, EfficientNetB2, SqueezeNet, VGG16, AlexNet, and LeNet-5, with different learning rates. This study indicates that our proposed approach outperforms these established models in terms of performance.

## 1. Introduction

Cancer is a serious illness that can spread to other areas of the human body and is known by uncontrolled cell development. According to statistics from the World Health Organization (WHO), an estimated 10 million deaths worldwide were reported due to cancer by the year 2020 [1]. However, these are estimates and may not fully reflect the actual incidence due to incomplete reporting systems, especially in underdeveloped countries. The tumors can be malignant (cancerous), with varying levels of aggressiveness, or benign (non-cancerous). The brain is the most important organ in humans, overall responsible for managing the body, and therefore, brain tumors need urgent attention. The incidence and survival of brain tumors are affected by their type, size, grade, and location, as well as the age of a patient and overall health condition [2]. To improve patient outcomes, early diagnosis, and adequate treatment are critical.

Medical imaging is an important tool for visualizing the internal structure of tumors. It includes magnetic resonance imaging (MRI), positron emission tomography (PET), and computed tomography (CT). Here, MRI is considered more appropriate than other medical imaging modalities, as it provides detailed images of soft tissues, organs, and internal structures without using harmful radiation. It also provides excellent contrast resolution, allowing for better differentiation between healthy and abnormal tissue. When MRI examination is required, professionals often use software to manually mark regions of interest (ROI) which is time-consuming and error-prone and might compromise diagnosis. It is critical to identify warning signals and seek medical attention as soon as possible because early treatment can save lives [3]. That is why automatic systems that extract MRI data have various benefits, including faster diagnosis and effective treatment, improved accuracy, consistency, cost-effectiveness, and time savings [4]. Automatic systems require less time and deliver superior performance compared to manual methods because they are accurate and adept at handling large amounts of medical data.

The emergence of artificial intelligence (AI) presents promising solutions in various applications [5,6,7,8] and especially for automating medical image analysis, thereby achieving faster and more accurate diagnoses [9]. Similarly, its sub-field, i.e., the DL techniques, has been widely used in radiology to extract robust features from MRI, PET, and CT scans. In the past, medical ML techniques have mostly concentrated on high-level or low-level features that are manually extracted during feature engineering. The low-level features capture the essential aspects of an image, whereas high-level features provide semantic information. However, methods that can combine both kinds of features are automatically becoming useful [10]. In the meanwhile, deep learning techniques like deep convolutional neural networks (DCNN) are helpful for automatically learning high-level features including tumor size, location, and surrounding tissue. These low-level and high-level features must be taken into account for reliable and accurate brain tumor classification. This study proposes a novel Dual DCNN for accurate brain tumor classification using MRIs. Our model addresses the challenge of comprehensive feature extraction by learning both low-level (i.e., texture, intensity) and high-level (i.e., tumor location, size) features directly from the MRI scans. This combined feature extraction capability allows the Dual DCNN to achieve high accuracy in classifying cancerous and non-cancerous brain tumors.

Our contributions are discussed in detail in upcoming sections. A brief summary is as below:Our proposed Dual DCNN model with denseNet121 and inceptionV3 has shown promising results. We observed significant improvements in various performance metrics especially the accuracy demonstrating the capability to accurately classify cancerous and non-cancerous MRI samples.We have implemented SOTA DL models, i.e., denseNet121, inceptionV3, resNet50, resNet34, resNet18, efficientNetB2, squeezeNet, VGG16, alexNet, leNet-5, and compared results with our methodology. We have highlighted the best performance of our approach.We have compared the performance of each SOTA DL model with different learning rates and identified the best learning rate for each model.We compared our approach with the latest research in cancer detection and classification and through benchmarking we found our proposed approach outperformed existing methods.

## 2. Related Work

In recent years, researchers have been utilizing DL techniques for the classification of brain tumors using MRI data. These studies utilize the power of DL algorithms, particularly DCNNs to extract relevant features from MRI scans and classify them into different tumor types. Scafuto et al. [11] used CNN for the classification and segmentation of MRI scans to examine glioma diagnosis studies. The authors collected 77 academic articles and emphasized the requirement of early and accurate detection of tumors for glioma patients’ survival. Their study aimed to implement a grading system for glioma brain tumors using DL and MRI. The authors used a dataset of 259 patients with glioma brain tumors. The researchers used a ConvNet model to divide the tumors into different levels. With an overall accuracy of 91.5%, the results showed that their proposed method achieved better accuracy for brain tumors caused by gliomas. Additionally, this study highlighted that their approach performed better as compared to the other methods. They suggested the integration of multi-modal data for more comprehensive analysis. Similarly, Fathima et al. [12] provide an overview of the evolution of data science disciplines and compare pre-processing, ML, and DL methods for brain tumor classification. Their methodology involved a review of existing techniques by highlighting the importance of early detection of brain tumors and early treatment. The authors discussed the advantages and limitations of different ML and DL techniques for brain tumor classification. They also compared the performance of different models and algorithms used in the literature. This study concluded that DL methods, particularly CNNs, have shown better results in classifying brain tumors. The authors also discussed the limitations of using deep learning methods, i.e., the need for large datasets for fine-tuning and the interpretability of the approach. The future direction of their study involves the development DL model with the integration of multi-modal data for more comprehensive analysis.

Several recent studies have also explored the use of fine-tuned approaches with complex datasets for the detection of brain tumors. Ghosal et al. [13] implemented a CNN-based SE-ResNet-101 that was fine-tuned for classifying brain tumors into malignant, gliomas, and pituitary tumors. They applied different pre-processing procedures, such as normalization, zero-centering, ROI segmentation, and augmentation of data like rotation, scaling, zooming, etc. Utilizing T1-weighted contrast-enhanced (T1W-CE) MRI data, their SE-ResNet-101 model obtained an accuracy of 93.83% with data augmentation and 89.93% without an augmentation approach. A three-step method was suggested by Nawaz et al. [14] which involved annotating interest regions, utilizing a customized CornerNet to extract deep features, and using a one-stage detector for tumor classification. Their accuracy on the T1W-CE MRI dataset was 98.8%. This approach provides an affordable solution because of the one-stage object identification framework provided by CornerNet.

The potential of hybrid approaches combining ML and DL has also been investigated. Mohammed et al. [15] offered four suggested systems for the early detection of brain cancers, each of which combines hybrid learning techniques. The first system uses hybrid features obtained from grey-level co-occurrence matrix (GLCM), discrete wavelet transform (DWT), and local binary pattern (LBP) methods to integrate artificial neural network and feedforward neural network techniques. The second method, a DL approach, achieved better results in differentiating between different types of brain cancers by using ResNet-50 and GoogLeNet models for classification. The third system shows the best results in tumor classification by utilizing a hybrid approach that combines support vector machines (SVMs) with CNN. The fourth system suggested uses a hybrid technique that combines DWT, GLCM, and LBP with ResNet-50 and GoogLeNet algorithms to obtain a more comprehensive analysis of brain tumor images. Similarly, Nassar et al. [16] used a hybrid deep learning strategy to build an effective automated method for classifying brain cancers. To take advantage of the combined abilities of numerous models and provide promising outcomes, the suggested method is based on the output of five distinct models. With an overall accuracy of 99.31%, their study’s T1-weighted brain MR images achieved superior results. However, Saha et al. [17] proposed an improved method that uses DL and a collection of ML algorithms to categorize types of brain tumors from MRI. The BCM-VEMT method, which may be classified into four separate classes, i.e., glioma, meningioma, pituitary, and non-cancerous (normal), is presented in their work. To extract features from the MRI, a CNN is developed and then input into multi-class machine learning classifiers. By combining each machine learning classifier, a weighted average ensemble of classifiers is utilized to improve performance. A total of 3787 MRI images from each of the four classes composed the dataset used in their study. With an overall accuracy of 98.42%, their system obtained high precision for each class: 97.90% for glioma, 98.94% for meningioma, 98.00% for normal, and 98.92% for pituitary. The need to have an image-processing computer-aided diagnosis system that can accurately classify different forms of brain tumors was highlighted by this study.

Researchers also used the transfer-learning approach to overcome the data limitation. Mahmud Badhon et al. [18] work on enhancing the DL transfer-learning technique to increase the accuracy of brain cancer diagnosis. Using a DL pre-trained network, VGG16, this study assesses individual performance in classifications with various evaluation metrics. The authors extracted characteristics from the image dataset using the convolutional block and dense layers of the VGG16 architecture. These features were then sent to machine learning classifiers, which calculated the final classification result. The research achieved significant accuracy using the DL-embedded ML technique. Al Rub et al. [19] use the DL model for the classification of hydrocephalus in brain computed tomography (CT) medical images. They created a precise and non-invasive method for brain hydrocephalus diagnosis. A dataset of 500 brain CT scans, split into training and testing sets during experimentation. The images were divided into groups for hydrocephalus and normal and achieved an overall accuracy of 96.8%, the results showed that their approach achieved better accuracy in detecting hydrocephalus in brain CT images. Additionally, this study demonstrated that the suggested approach performed well as compared to other techniques. A study by Mehnatkesh et al. [20] focuses on classifying brain cancers in MRI scans using deep learning algorithms. To save radiologists time by eliminating the need to examine several images to make an accurate diagnosis, the aim of their work was to create an effective automated method for classifying brain tumors. A total of 3064 T1w-CE brain MRI scans were used from 233 patients. To achieve promising results, they proposed a deep residual learning framework that combines the capabilities of multiple models. With an overall accuracy of 99.31%, the results demonstrated that their proposed approach showed better performance in classifying brain cancers in MRI scans.

The remaining part of the article is divided into three sections. Section 3 describes the details of the methodology used in this study, Section 4 provides a detailed explanation of results, and Section 5 includes a conclusion with possible future direction.

## 3. Methodology

DL models with DCNNs are particularly well suited for the extraction of robust and important features from images which are essential to understanding the complex structure of MRIs. Our research uses this advantage by employing two well-performed DL models, i.e., InceptionV3 and DenseNet121. The individual performance of both networks is explained in the results section of the paper. By merging the feature representations from both DL models, the fusion technique used in our dual Dual DCNN model combines the strengths of both models. It captures more varied and informative features by integrating the feature maps from both networks and training extra fully connected layers, potentially leading to improved performance when compared to using each network alone. The overview of our approach is shown in Figure 1, whereas details are provided later in the subsection.

### 3.1. Dataset Description

Our experiments utilized a Kaggle dataset, i.e., Br35H [21] comprising a total of 3000 MRI images where 1500 are tumors and 1500 without tumors samples. Each image in the dataset was 256 × 256 pixels in dimension. The images were labeled as either “yes” for cancerous and “no” for non-cancerous samples. The training and validation details of both classes are shown in Table 1. The dataset was split between cancerous and non-cancerous images. We combined these images and randomly divided them into 80% and 20% for training and validation, respectively, using splitfolder library which divides the dataset randomly. We assigned labels as “0” for normal cases and “1” for brain tumors in the proposed approach and SOTA DL models. Figure 2 shows MRI samples taken from the dataset, i.e., cancerous and non-cancerous images.

### 3.2. Dual DCNN Model

Our dual DCNN (DDCNN) model integrates features obtained from DenseNet121 and InceptionV3, i.e., two pre-trained DL model architectures. Combining their respective strengths in feature extraction, the fusion method combines the learned representations from both networks. We used the torchvision library to load the InceptionV3 and DenseNet121 models that have already been trained and perform well in feature extraction, they are often used for image classification tasks [22,23,24] for various applications. To benefit from their pre-trained weights for feature extraction, we froze the parameters of both InceptionV3 and DenseNet121 during training. This ensures that the pre-trained weights are not retained by scratch and only the fully connected layers are trained. Next, we add identity layers to each model in place of the classifier layers. By removing the models’ last classification layers of each network, we extract features immediately from the convolutional layers. We concatenate the feature maps from the DenseNet121 and InceptionV3 models along the channel dimension in the fusion model. As a result, a combined feature tensor which includes data from both networks is created. In order to process the resulting feature tensor, we add fully connected layers to the fusion model after the concatenation phase. The retrieved features from both networks are combined and refined in these layers. Lastly, we use an optimization approach (such as the Adam optimizer) and a specified loss function, i.e., binary-entropy loss to train the fusion model. During training, the system effectively combines the features from InceptionV3 and DenseNet121 to classify the cancer and non-cancer MRI samples.

#### 3.2.1. Preprocessing

The image input layer pre-processes the dataset before passing it into the DDCNN model. It typically involves resizing the images to a uniform size, i.e., 229 × 229 and normalizing the pixel values to ensure consistency and convergence during training. The normalization process is described as:(1)$$x_{\mathrm{norm}\;}=\frac{x-\mu }{\sigma }$$
where *x* represents the pixel value, μ is the mean pixel value across the dataset, and σ is the standard deviation of the pixel values. It ensures that the pixel values have a zero mean and unit variance.

#### 3.2.2. Features Extraction

This study uses denseNet121 and inceptionV3 DL models to extract the features. DenseNet core innovation is its dense connection pattern where all layers are feed-forwardly coupled to all other layers. A dense connection lowers the number of parameters, promotes gradient flow throughout the network, and makes feature reuse easier. This extensive connection mitigates the vanishing gradient issue that deep neural networks often encounter and allows optimal parameter reuse. Additionally, DenseNet architectures can be improved for feature extractors for a variety of image recognition tasks [25,26,27] because they have already been pre-trained on large datasets, i.e., ImageNet. By removing the fully connected layers and substituting identity functions, the DenseNet121 model is used in the given study as a feature extractor which turns it into a feature extraction module. Concatenating the feature maps from all preceding layers serves as the input for each layer, resulting in the following input tensor size:(2)$$\left(C_{l}+C_{l-1}+\ldots +C_{0},H,W\right)$$
where $C_{l}$ is the number of the current layer’s channel output. *H* and *W* are the height and width of the feature maps.

The second model employed in this approach is InceptionV3, a deep learning architecture renowned for its exceptional capability in various image recognition tasks. It consists of many layers with different convolutional filters, such as max and average pooling layers, and convolutions of sizes 1 × 1, 3 × 3, and 5 × 5. To improve the model performance and adaptability, the architecture also includes batch normalization, dropout, and other methods. To minimize computational complexity and maintain representational ability, it utilizes a factorized convolution approach. Similar to DenseNet121, InceptionV3 is a feature extractor that has been pre-trained on large datasets like ImageNet. The InceptionV3 model is utilized in this study in a similar way as DenseNet121, i.e., the convolutional base remains for feature extraction but the fully connected layers are removed. The collected features from DenseNet121 and InceptionV3 are concatenated and sent through additional fully connected layers for final classification. A more thorough representation of the input data is made possible by combining features from these two DL models which enhances the performance on tasks including robust high and low features’ extraction and accurate classification.

This study also uses GlobalMaxPooling2D. This operation is typically used as a means of spatial downsampling to reduce the dimensionality while preserving the most salient (or low-level but important) information. In the DDCNN model, GlobalMaxPooling2D is set up after the convolutional layers in each DL model, i.e., DenseNet121 and InceptionV3. By choosing the largest activation value from each feature map, this pooling method successfully highlights the most important characteristics in the input data. The GlobalMaxPooling2D operation is applied along the spatial dimensions of the feature maps. Considering an input tensor *X* with dimensions (C,H,W), where *H* and *W* stand for the feature maps’ height and width, *C* indicates the number of channels. GlobalMaxPooling2D operation can be expressed as follows:(3)$$\;\mathrm{GlobalMaxPooling}\;2D\left(X\right)=\max\left(X_{c,h,w}\right)\;\mathrm{for}\;h=1,\ldots ,H\;\mathrm{and}\;w=1,\ldots ,W$$

Here, $X_{c,h,w}$ is the activation value at channel *C* and spatial position (h,w). This operation results in a tensor with dimensions (C,1,1).

The utilization of GlobalMaxPooling2D in the DDCNN model contributes to model efficiency and robustness by reducing the number of parameters and preventing overfitting. Moreover, this helps achieve translational invariance that strengthens the model’s resistance to shifts or translations in space in the input data. The difference between the max pooling and global max pooling is shown in Figure 3 where the maximum value from the whole region of interest is selected for global pooling in 6 × 6 matrix.

#### 3.2.3. Fully Connected Layers

The features extracted by DenseNet121 and InceptionV3 are concatenated along the channel dimension before being fed into the fully connected layers. Let $x_{dense}$ be the features that DenseNet121 collected and $\left(N,C_{dense},H_{dense},W_{dense}\right)$ are dimensions of these features, similarly, $x_{inception}$ be the features of incpetionV3 and $\left(N,C_{inception},H_{inception},W_{inception}\right)$ are dimension of it. The resultant concatenation feature vector would be as:(4)$$x_{concat}=\left(N,C_{\mathrm{dense}\;}+C_{\mathrm{inception}\;},H_{\mathrm{dense}\;},W_{\mathrm{dense}\;}\right)$$

In this study, the denseNet121 has 1024 logits 1D feature vectors and inceptionV3 has 2048 logits 1D feature vectors. The resultant concatenated 1D feature vector was 3072. These are important features obtained from both pre-trained DL models. These features’ tensorsare then passed through a series of fully connected layers to perform classification. These fully connected layers consist of linear transformations followed by the Rectified Linear Unit (ReLU). Let $W_{i}$ be the weight matrix of the *i*-th fully connected layer and $b_{i}$ the bias vector. The following formula is used to get the *i*-th fully connected layer output:(5)$$x_{i}=\mathrm{ReLU}\left(W_{i}x_{\mathrm{reshaped}\;}+b_{i}\right)$$

There are in total five fully connected layers with 3072,512,256,128, and 64 neurons belonging to FC1,FC2,FC3,FC4, and FC5, respectively. These layers learn the features and are fine-tuned with the Br35H dataset and hence provide better performance.

#### 3.2.4. Output Layer

The last layer of the Dual DCNN model applies a softmax activation function to transform the raw output scores into probability distributions over the output classes. Let $x_{\mathrm{final}\;}$ denote the final output vector before softmax, and $\hat{y}$ denote the predicted probability of cancer and non-cancer class. The softmax function will be defined as:(6)$$\hat{y}=\mathrm{softmax}\left(x_{\mathrm{final}\;}\right)$$
where
(7)$$\mathrm{softmax}{\left(x_{\mathrm{final}\;}\right)}_{i}=\frac{e^{x_{\mathrm{final}\;,i}}}{\sum_{\mathrm{classes}\;}e^{x_{\mathrm{final},j}}}$$

Through fully connected layers, the DDCNN model integrates the features of denseNet121 and inceptionV3 by utilizing the matching knowledge of both models, potentially increasing overall performance.

### 3.3. SOTA DL Models

This study includes a comparative analysis of DDCNN models with ten well-known DL models, i.e., LeNet-5, AlexNet, VGG-16, SqueezeNet, EfficientNetB2, ResNet18, ResNet34, ResNet50, InceptionV3, and DenseNet121. By evaluating the performance of the DDCNN alongside these widely recognized DL architectures, this study aims to provide insights into the effectiveness of the proposed model compared to existing state-of-the-art approaches. This study uses these DL models with transfer learning capitalizing on their knowledge gained from extensive training on large datasets (e.g., ImageNet) significantly reducing training time and computational resources. Additionally, this allows faster convergence and enhanced generalization which is valuable when dealing with limited medical image datasets such as those encountered in brain tumor classification tasks using MRI.

### 3.4. Training Parameters

This study used the same training parameters for SOTA DL models and the DDCNN model for the classification of cancerous and non-cancerous MRIs. We compared evaluation parameters with five different learning rates and used 128 images as a batch size that shuffled at every epoch with a total of fifty epochs as shown in Table 2. We used the same hyperparameter settings for all SOTA DL models and the DDCNN approach for a fair comparison of performance against each model. The binary cross entropy (BCE) is used with an Adam optimizer that calculates the difference between the actual binary labels and the expected probability distribution as:(8)$$BCE_{Loss}=-\frac{1}{N}\sum_{i=1}^{N}\left(y_{i}\cdot \log\left(p_{i}\right)+\left(1-y_{i}\right)\cdot \log\left(1-p_{i}\right)\right)$$
where *N* is the total number of samples, $y_{i}$ is the true label (0 or 1) of the *i*th sample, and $p_{i}$ is the predicted probability that the *i*th sample belongs to class 1.

## 4. Experimentation and Results Discussion

### 4.1. Experimental Setup

This study utilized a 15-inch Mackbook Proequipped with a dedicated GPU, 16 cores, 19 high-performance CPU cores, and an M2 chip with a processing speed of up to 4.0 GHz for efficient performance. To facilitate the development and comparison of different deep learning models, a separate virtual environment was created. We installed all the required libraries and packages in it. The Keras library within the tenserflow framework was used for VGG-16, LeNet-5, and AlexNet, whereas the PyTorch framework was used for building SqueezeNet, EfficientNetB2, ResNet18, ResNet34, ResNet50, InceptionV3, DenseNet121, and the proposed DDCCN model.

### 4.2. Evaluation Protocol

The f1-score, recall, accuracy, and precision were considered for the evaluation of the binary classification of DL models. The confusion matrix was also computed. True positives (TPs), true negatives (TNs), false positives (FPs), and false negatives (FNs) are its four coefficients. Accuracy is defined as the proportion of correctly expected cases among all instances. Precision is defined as the proportion of true positives among all positive predictions. The percentage of true positives predicted among all actual positive outcomes is called recall. It is also sometimes referred to as sensitivity or true positive rate. The harmonic mean of recall and precision provides the F1-score.
(9)$$\;\mathrm{Accuracy}\;=\frac{TP+TN}{TP+TN+FP+FN}$$
(10)$$\;\mathrm{Precision}\;=\frac{TP}{TP+FP}$$
(11)$$\;\mathrm{Recall}\;=\frac{TP}{TP+FN}$$
(12)$$F1\_\mathrm{score}\;=2\times \frac{\;\mathrm{Precision}\;\times \;\mathrm{Recall}\;}{\;\mathrm{Precision}\;+\;\mathrm{Recall}\;}$$

### 4.3. Results Discussion

This section analyzes the results acquired using the DDCNN model with the impact of different learning rates. It provides a detailed breakdown of the actual versus predicted classifications for each class of samples.

Figure 4 shows the classwise actual and predicted samples obtained using five different learning rates. Figure 4e shows that the highest learning rate, i.e., 0.1, only predicted actual positive samples correctly and did not classify the actual negative samples. In other words, this learning rate classified all 600 samples as cancerous. It was observed that the best results were obtained using a 0.0001 learning rate (i.e., Figure 4b) whereas only 12 actual negative samples were misclassified out of a total of 600 samples. Other learning rates with respective confusion matrices were also shown with actual and predicted samples where a 0.00001 learning rate performed well but misclassified 15 (i.e., Figure 4a) cancerous and non-cancerous MRI samples.

Figure 5 shows the training and validation accuracy and loss of the Dual DCNN model using the best 0.0001 learning rate. Both the validation accuracy and training accuracy increase as the number of epochs (training iterations) progresses. However, around epoch 30, both curves seem to plateau. This suggests that the model’s performance has reached its optimal point. It is essential to balance high accuracy on the training data and generalization to unseen validation data. The loss initially decreases sharply. After approximately 10 epochs, the loss stabilizes. This indicates that the model effectively learns during the initial epochs but may not improve significantly beyond that point. Monitoring loss helps prevent overfitting and ensures the model generalizes well.

The evaluation parameters against different models trained at best learning rates are shown in Table 3. With 99% accuracy and precision, the DDCNN model performed best, demonstrating its ability to correctly classify nearly all cases with extremely few false positives. Additionally, it obtained a high recall of 98%, which indicates that the majority of positive samples were successfully identified. As a result, its F1-score, a measurement for both recall and precision, was 99%, also outstanding. DenseNet121 achieved good performance with an accuracy of 97% and properly recognized a high majority of instances. Additionally, its precision and recall were both 97%, indicating that it performed fairly in detecting positive occurrences with few false positives.

With accuracy ranging from 95% to 96%, the InceptionV3, ResNet50, ResNet34, ResNet18, and EfficientNetB2 models performed similarly. Although their recall and precision scores were likewise quite satisfactory, there were slight variations across the models. For example, InceptionV3 got a lower F1-score than ResNet models. Models like SqueezeNet, VGG-16, AlexNet, and LeNet-5 performed lower than the previously mentioned ones. Although their levels of accuracy were still fair, they performed far worse in terms of precision, recall, and F1-scores. Particularly, AlexNet and LeNet-5 performed less as they had the lowest results across all criteria.

Figure 6 shows the confusion matrices of ten SOTA DL models used in this study. It is the classwise correct and incorrect prediction of two classes that allow a better understanding of the performance of individual models against best learning rates. Figure 6a shows the confusion matrix of DenseNet121 DL model which is dominating with learning rate of 0.001. It also shows the highest evaluation parameters using SOTA DL models. Whereas, Figure 6b–h shows the better performance of InceptionV3, ResNet50, ResNet34, ResNet18, EfficientB2, SqueezeNet and VGG-16, respectively. These DL models are showing performance matrices greater than 90% which are considered as acceptable in binary classification problem. However, Figure 6i,j shows the evaluation parameters of AlexNet and LeNet-5 using best learning rates. These both are not performing well.

Similarly, Figure 7 shows the accuracy, precision, recall, and f1-scores against all models used in this study which can be seen from Table 3 as well. Figure 7a shows the accuracy which is ratio of correctly predicted samples to the overall samples. Figure 7b–d shows the percentage of precision, recall, and F1, respectively against all DL models used in this study.

Table 4 shows the classwise performance matrices (accuracy, precision, recall, and f1-scores). The DDCNN model achieved a high accuracy of 99%, precision of 99%, recall of 98%, and f1-score of 99% for non-cancer (class 0). This suggests that with extremely few false positives, it accurately classified nearly all cases of non-cancerous samples. It also performed better for cancerous cases (Class 1), showing the perfect classification of cancer with 99% accuracy, precision, recall, and f1-score. With all metrics ranging around 97%, DenseNet121 performed well for class 0. Its accuracy, precision, recall, and f1-score were lower than those of the DDCNN model. With accuracy, precision, recall, and f1-score all ranging around 96% for cancerous cases, the performance remained unchanged. This suggests that the cases of cancer were accurately classified.

InceptionV3, ResNet50, ResNet34, ResNet18, EfficientNetB2, SqueezeNet, and VGG-16 models have generally similar trends across both classes, with slightly varying performance metrics. With somewhat different performance criteria, the InceptionV3, ResNet50, ResNet34, ResNet18, EfficientNetB2, SqueezeNet, and VGG-16 models largely showed comparable patterns in both classes. Class 0 accuracy was between 93% and 97%, with slightly lower but still good precision, recall, and F1-score. With accuracy ranging from 91% to 95% and modest changes in precision, recall, and F1-score, class 1 performance was slightly lower than class 0. The AlexNet and LeNet-5 models underperformed as compared to other models. For class 0, precision, recall, and F1-score were similarly comparatively lower, whereas accuracy ranged from 79% to 90%. With accuracy ranging from 64% to 80% and precision, recall, and F1-score showing a similar decreasing trend, class 1 performance decreased even more.

In conclusion, the DDCNN model outperformed compared to the others in both classes, achieving almost perfect recall, accuracy, precision, and F1-score. However, AlexNet and LeNet-5 performed relatively worse while taking into account all evaluation parameters.

Table 5 presents the influence of five different learning rates on the DDCNN model and several SOTA DL models trained with five different learning rates. The Dual DCNN performance demonstrates sensitivity to the learning rate. The best results were obtained when training at a learning rate of 0.0001, achieving 99%, 99%, 98%, and 99% for accuracy, precision, recall, and F1-score, respectively, whereas performance decreased from 0.0001 to 0.00001 when the learning rate dropped. However, it remained quite satisfactory, with an accuracy of 98% and other metrics over 97%. Its lowest performance was for a higher learning rate as shown in the table. Similar to the Dual DCNN, models like InceptionV3, ResNets, EfficientNetB2, DenseNet121, SqueezeNet, and VGG-16 demonstrated inconsistent performance at different learning rates. Lower learning rates were often associated with better performance. For instance, DenseNet121 and ResNet18 both achieved their maximum accuracy of 93% and 96%, respectively, with a learning rate of 0.001 and 0.01, respectively.

Across all learning rates, AlexNet and LeNet-5 underperformed compared to the other models in terms of accuracy, precision, recall, and f1-score. For instance, for a given learning rate of 0.0001, AlexNet’s maximum accuracy was 85%, whereas LeNet-5’s maximum accuracy was 71% at 0.001.

In conclusion, the findings highlight the crucial role of hyperparameter tuning, particularly learning rate selection, in optimizing DL model performance. Performance was generally better at lower learning rates, whereas the ideal learning rate varied based on the particular model architecture. Additionally, several models showed consistency in their performance across a variety of learning rates, whereas other models showed variability. These results highlight how important it is to adjust hyperparameters, including the learning rate, in order to maximize the DL model’s performance.

### 4.4. Comparison of SOTA versus DDCNN Model

Table 6 shows the Dual DCNN’s exceptional performance compared to recent studies employing various DL models validated using MRI images having cancerous (and its type) and non-cancerous classes. In these studies, our DDCNN model achieves an accuracy of 99% in classifying cancerous and non-cancerous MRI scans, surpassing the highest accuracy reported in the reviewed literature. Even though other techniques, i.e., optimized EfficientNetB2 and MSGGAN achieve high accuracy above 98%, our DDCNN APPROACH still exceeds them. Moreover, transfer-learning-based models like DenseNet201 and GoogleNet, together with innovative architectures like Res-BRNet and Self CNN also attain competitive accuracies between 97% and 98%, they are unable to match our performance. These findings suggest that the Dual DCNN represents a superior performance in deep learning for MRI-based cancer and non-cancer classification tasks.

## 5. Conclusions

Detecting brain tumors at an early stage is very crucial as it increases the chances of patient survival through effective treatment. The manual diagnosis by doctors is slow and subject to inter-observer variations, especially with the increasing number of new cases reported on a daily basis. Hence, there is an urgent need for rapid analysis of large medical data to enable early tumor diagnosis. Here, AI with deep learning methods are reliable and efficient approach because it can handle complex medical data like MRI scans more precisely. This study introduces the Dual DCNN (DDCNN) model, which utilizes two high-performing DL models that achieve better performance than SOTA models. It compares results with various well-known DL models, including DenseNet121, InceptionV3, ResNet50, ResNet34, ResNet18, EfficientNetB2, SqueezeNet, VGG16, AlexNet, and LeNet-5. Furthermore, this study compares the proposed Dual DCNN approach with recent research and demonstrates better performance.

Future efforts will focus on the prediction of glioma grade which plays an essential role in guiding treatment decisions. Transformers, which were introduced recently and became popular, can be instigated and compared with Dual DCNN approach.

## Figures and Tables

**Figure 1 bioengineering-11-00410-f001:**
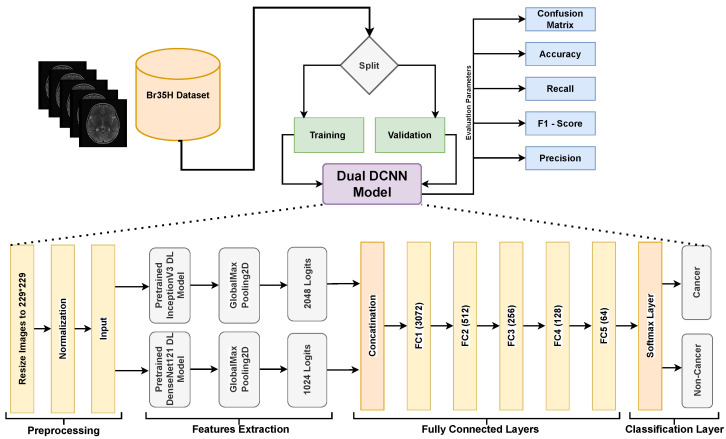
A block diagram for the comprehensive overview of our proposed approach to understand the dual DCNN architecture and workflow.

**Figure 2 bioengineering-11-00410-f002:**
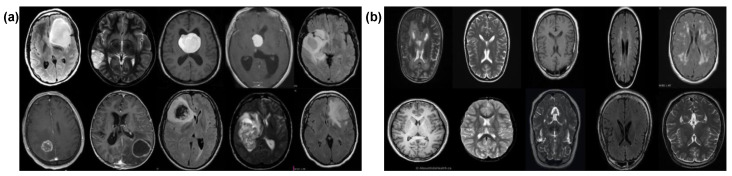
MRI samples: (**a**) cancerous, (**b**) non-cancerous.

**Figure 3 bioengineering-11-00410-f003:**
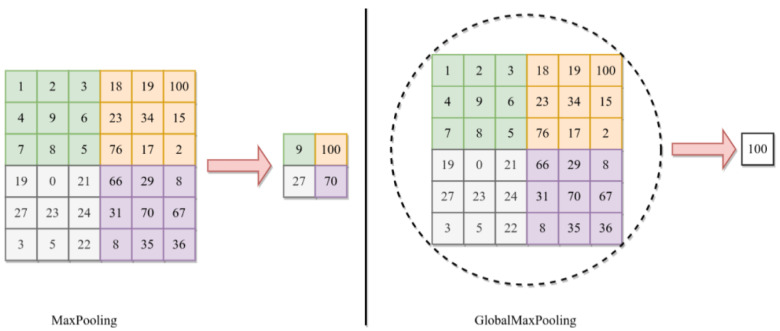
The basic difference between the GlobalMaxPooling and MaxPooling.

**Figure 4 bioengineering-11-00410-f004:**
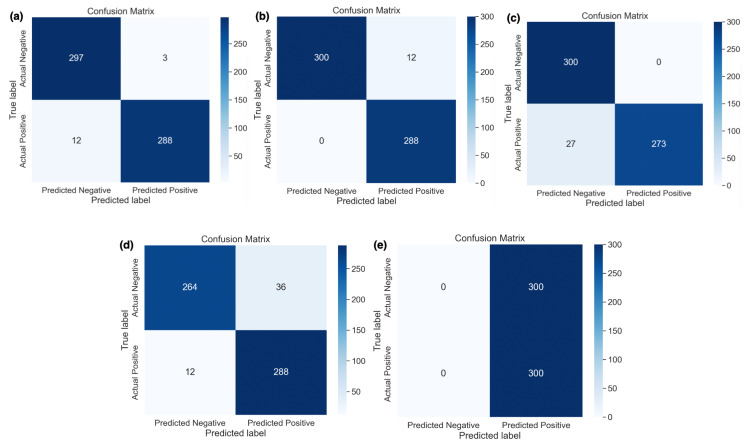
Confusion matrix of DDCNN model with different learning rates (**a**) 0.00001, (**b**) 0.0001, (**c**) 0.001, (**d**) 0.01, and (**e**) 0.1.

**Figure 5 bioengineering-11-00410-f005:**
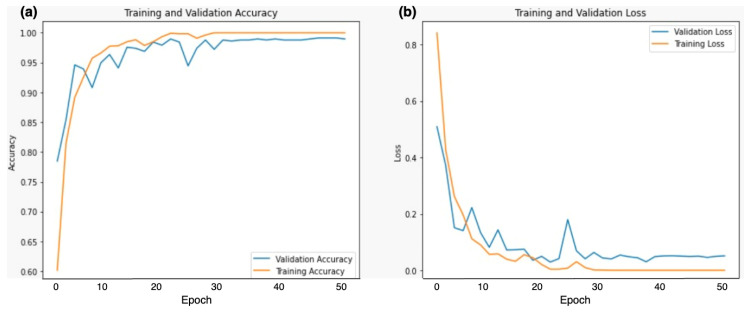
Training and validation (**a**) accuracy (**b**) loss of DDCNN model using 0.0001 learning rate.

**Figure 6 bioengineering-11-00410-f006:**
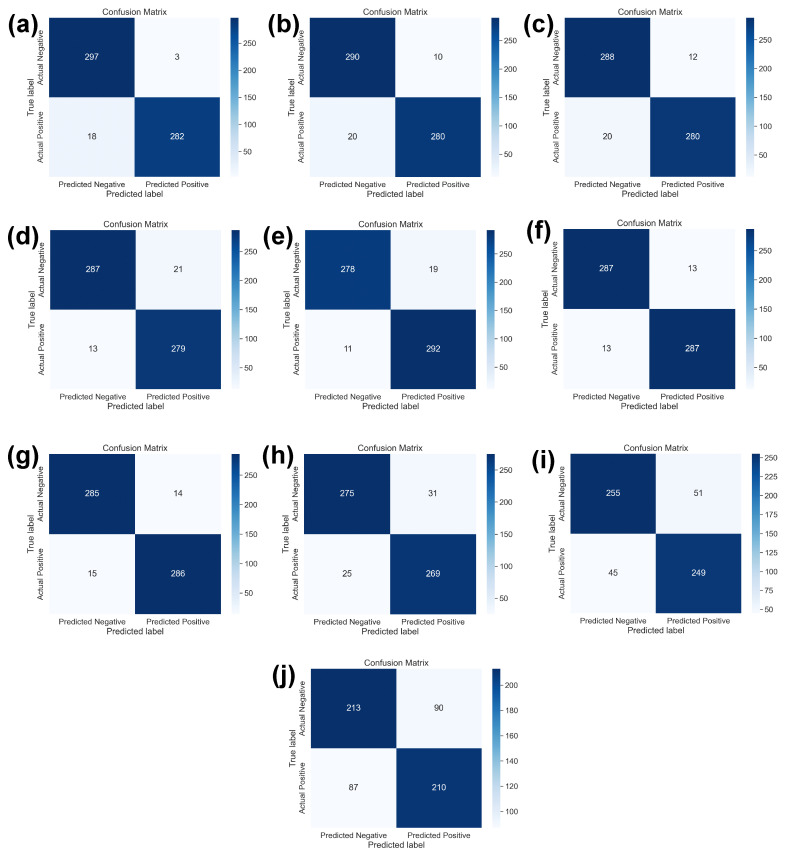
The confusion matrix of all ten SOTA DL models with respect to the best learning rates as mentioned in Table 3. (**a**) DenseNet121 with 0.0001 learning rate, (**b**) InceptionV3 with 0.001 learning rate, (**c**) ResNet50 with 0.00001 learning rate, (**d**) ResNet34 with 0.00001 learning rate, (**e**) ResNet18 with 0.0001 learning rate, (**f**) EfficientNetB2 with 0.00001 learning rate, (**g**) SqueezeNet with 0.0001 learning rate, (**h**) VGG16 with 0.0001 learning rate, (**i**) AlexNet with 0.0001 learning rate, and (**j**) LeNet-5 with 0.001 learning rate.

**Figure 7 bioengineering-11-00410-f007:**
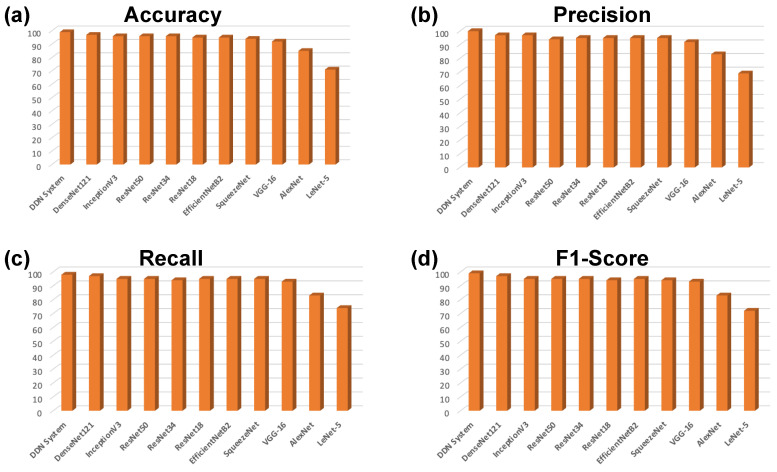
Evaluation parameters for DDCNN and SOTA DL models with respect to learning rates provided in Table 3. The (**a**) Accuracy, (**b**) Precision, (**c**) Recall, and (**d**) F1-Scores bar-graphs for all DL models used in the study.

**Table 1 bioengineering-11-00410-t001:** Dataset split into training and validation and respective class label.

Class	Images	Train	Validation	Label
Cancerous	1500	1200	300	1 “Yes”
Non-Cancerous	1500	1200	300	0 “No”
**Total**	**3000**	**2400**	**600**	

**Table 2 bioengineering-11-00410-t002:** Training parameters used in SOTA DL and DDCNN model.

Sr. No	Parameters	Value
1	Learning Rates	0.1, 0.01, 0.001, 0.0001, and 0.00001
2	Batch Size	128
3	Number of Epochs	50
4	Loss Function	Adam Optimizer With Binary Cross-Entropy
5	Shuffle	Every Epoch

**Table 3 bioengineering-11-00410-t003:** Evaluation parameters via best learning rates.

Model	Learning Rate	Accuracy	Precision	Recall	F1-Score
DDCNN	0.0001	**99**	**99**	**98**	**99**
DenseNet121	0.001	97	97	97	97
InceptionV3	0.01	96	97	95	95
ResNet50	0.00001	96	94	95	95
ResNet34	0.00001	96	95	94	95
ResNet18	0.0001	95	94	95	94
EfficinetNetB2	0.00001	95	95	95	95
SqueezeNet	0.0001	94	95	95	94
VGG-16	0.0001	92	92	93	93
AlexNet	0.0001	85	83	83	83
LeNet-5	0.001	71	69	74	72

**Table 4 bioengineering-11-00410-t004:** Classwise evaluation parameters of DDCNN model and SOTA DCNN models.

Model	Learning Rate	Class	Accuracy	Precision	Recall	F1-Score
DDCNN model	0.0001	0	**99**	**99**	**98**	**99**
		1	**99**	**99**	**99**	**99**
DenseNet121	0.001	0	98	97	97	97
		1	96	96	96	96
InceptionV3	0.01	0	97	96	95	95
		1	95	96	96	94
ResNet50	0.00001	0	97	97	96	95
		1	95	96	94	95
ResNet34	0.00001	0	97	96	95	96
		1	95	95	93	94
ResNet18	0.0001	0	96	94	96	94
		1	94	94	94	95
EfficinetNetB2	0.00001	0	96	95	95	95
		1	94	94	94	95
SqueezeNet	0.0001	0	95	96	95	94
		1	93	94	94	94
VGG-16	0.0001	0	93	92	93	93
		1	91	92	92	93
AlexNet	0.0001	0	90	84	81	84
		1	80	81	80	81
LeNet-5	0.001	0	79	75	78	79
		1	64	61	66	68

**Table 5 bioengineering-11-00410-t005:** Evaluation parameters of dual DCNN models and SOTA DL models with respect to five different learning rates.

Model	Learning Rate	Accuracy	Precision	Recall	F1-Score
**DDCNN Model**	**0.1**	50	25	50	33
	**0.01**	92	92	92	92
	**0.001**	95	96	96	95
	**0.0001**	**99**	**99**	**98**	**99**
	**0.00001**	98	98	98	98
**DenseNet121**	**0.1**	70	71	71	70
	**0.01**	68	71	68	66
	**0.001**	**97**	**97**	**97**	**97**
	**0.0001**	95	96	96	95
	**0.00001**	83	86	83	83
**InceptionV3**	**0.1**	48	46	48	42
	**0.01**	**96**	**97**	**95**	**95**
	**0.001**	94	95	94	94
	**0.0001**	74	78	74	74
	**0.00001**	50	50	50	50
**ResNet50**	**0.1**	67	67	67	66
	**0.01**	93	93	93	93
	**0.001**	95	95	96	95
	**0.0001**	77	79	77	78
	**0.00001**	**96**	**94**	**95**	**95**
**ResNet34**	**0.1**	68	68	68	67
	**0.01**	94	94	94	93
	**0.001**	95	94	95	94
	**0.0001**	67	69	67	66
	**0.00001**	**96**	**95**	**94**	**95**
**ResNet18**	**0.1**	58	74	58	50
	**0.01**	66	80	66	61
	**0.001**	94	94	94	93
	**0.0001**	**95**	**94**	**95**	**94**
	**0.00001**	93	94	93	93
**EfficinetNetB2**	**0.1**	67	66	66	65
	**0.01**	85	87	85	85
	**0.001**	89	91	89	89
	**0.0001**	93	94	93	94
	**0.00001**	**95**	**95**	**95**	**95**
**SqueezeNet**	**0.1**	50	25	50	33
	**0.01**	60	62	62	61
	**0.001**	91	92	91	91
	**0.0001**	**94**	**95**	**95**	**94**
	**0.00001**	90	90	90	90
**VGG-16**	**0.1**	50	25	50	33
	**0.01**	88	78	84	80
	**0.001**	91	92	91	91
	**0.0001**	**92**	**92**	**93**	**93**
	**0.00001**	90	90	90	90
**AlexNet**	**0.1**	50	25	50	33
	**0.01**	60	55	58	54
	**0.001**	83	83	83	83
	**0.0001**	**85**	**83**	**83**	**83**
	**0.00001**	84	83	82	82
**LeNet-5**	**0.1**	54	29	54	37
	**0.01**	61	64	59	61
	**0.001**	**71**	**69**	**74**	**72**
	**0.0001**	67	66	66	65
	**0.00001**	70	69	69	68

**Table 6 bioengineering-11-00410-t006:** Quantitative analysis of DDCNN model and latest related research.

Research	Methodology	Model	Accuracy
In [28]	CNN-Fine Tuned	EfficientNetB2	98.86%
In [29],	Generative Adversarial Network (GAN)	MSGGAN	98.57%
In [30],	DCNN-Transfer Learning	DenseNet201	98.22%
In [31],	CNN-Transfer Learning	GoogleNet	98%
In [32],	CNN-Novel	Self CNN	97.3%
In [33],	CNN-Cross Validation	Self CNN	96.56%
In [34],	Neural Network (NN)	Self NN	95.86%
In [35],	Siamese Neural Network (SNN)	MAC-CNN	92.8%
**Our Approach**	**Dual DCNN Model**	**DDCNN**	**99%**

## Data Availability

The dataset is publicly available and accessible via https://www.kaggle.com/datasets/viveknarayanuppala/br35h-binary (accessed on 13 August 2023).

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
