# Peer review of "Cancerous and Non-Cancerous MRI Classification Using Dual DCNN Approach"

_bioengineering, 2024, doi:10.3390/bioengineering11050410_

Round 1
Reviewer 1 Report
Comments and Suggestions for Authors
Authors need to incorporate the following suggestions.
1. The novelty of the proposed work need to be explained clearly in the abstract. Application of pretrained SOTA models cannot be considered as novelty.
2. Loss and accuracy convergence plots need to be presented with clear interpretation.
3. Authors need to justify on what basis, the DL models are selected. Why few models like EfficientB0, Darknet, Densenet169 etc were missed as part of the analysis?
4. Authors need to check the typos and phraseological errors. Ex. 4.3. Results Discussion
5. Results need to be validated with benchmark datasets like BRATS.
6. How the hyper parameters were set for the individual networks?
7. What is the train-validation-test distribution ratio? As the work reports 99% accuracy, did the authors change the distribution and analysed its effects?
8. Authors need to provide the details on the cross validation made.
Comments on the Quality of English LanguageModerate changes
Author Response
Dear Editor,
Please find the attached response file for Reviewer#1.
Regards,
Zubair Saeed

Reviewer 2 Report
Comments and Suggestions for Authors
The manuscript was well written for the classifying the brain tumors based on the 2D brain MRI images into cancer and non-cancerous type. This was performed using features selected from CNN based models, Pretrained Densenet121 and Inception V3. These combined features were used by the linear model to classify the 2D image slice into cancerous or non-cancerous type. The hyper-parameter tunning and the comprehensive results for all the comparing models were some of the highlights of the manuscript.
Few questions on the manuscript,
1. The dataset used for the project was “Br35h: Brain Tumor Detection 2020”. This dataset was a kaggle dataset. But the data citation, does not provide enough information of the dataset. The dataset lacks crucial information regarding the cites involved in curating the dataset, dataset origin and also the number of annotators used to annotate the dataset.
2. Also, Information regarding the dataset split with respect to subjects / patients is missing in the dataset. This information for the subject-wise dataset split was not provided in the manuscript.
3. Since the dataset contains only MRI 2D images of slices with tumor or non-tumor. Also, the information regarding the subjects for which the slices belong was not provided, there is possibility of dataset leakage between the train and the test set. If the images from the same subject are available in the train and test set, the CNN model would overfit on the images from the same subject. This in turn will be a case of data leakage between training and test set. The manuscript lacks in providing details for the information regarding the images and its corresponding subjects.
4. Also, the manuscript lacks the information on the dataset, specifically which type of MRI modalities were used for this study. Also, the pre-processing steps doesnot mention about any RF in-homogeneity correction for the MRI images. It would be good to know whether the RF in-homogeneity correction was performed on this dataset.
5. The models used for this analysis were DenseNet 121 and Inception V3, they are no longer the best performing model in the classification domain. Also, the models they are compared against are not the best performing model. The manuscript doesn’t compare the models to the recent best performing models in the classification challenge. None of the Transformer models were used for the comparison. It would be interesting to see the comparison between the current model and the transformer based (best performing) models.
6. Also, the manuscript has missing citations, line 231 in the manuscript.
7. The hyper-parameter tunning performed in the manuscript was great, it really highlights the part of selecting the optimum learning rate for the specific model.
8. Meta-analysis for the features that were being selected by the Inception V3 and the DenseNet 121 would be good to analyze and understand the feature selection process, this information was not provided in the manuscript.
Comments on the Quality of English LanguageEnglish language used was ok and require minor improvement.
Author Response
Dear Editor,
I hope you are doing well.
Please find the attached response file for Reviewer#2.
Regards,
Zubair Saeed

Round 2
Reviewer 1 Report
Comments and Suggestions for Authors
Suggestions were incorporated. Yet novelty can be emphasized properly.
Comments on the Quality of English LanguageMinor changes